# Application of Electrophoretic Deposition as an Advanced Technique of Inhibited Polymer Films Formation on Metals from Environmentally Safe Aqueous Solutions of Inhibited Formulations

**DOI:** 10.3390/ma16010019

**Published:** 2022-12-20

**Authors:** Natalia A. Shapagina, Vladimir V. Dushik

**Affiliations:** Frumkin Institute of Physical Chemistry and Electrochemistry, Russian Academy of Sciences, Leninsky Prospect 31-4, 119071 Moscow, Russia

**Keywords:** metals, electrophoretic deposition (EPD), cataphoretic (CPD) and anaphoretic deposition (APD), inhibited formulations (INFOR), organosilanes, corrosion inhibitors

## Abstract

The presented paper analyzes polymer films formed from aqueous solutions of organosilanes, corrosion inhibitors and their compositions. Methods of depositing inhibited films on metal samples, such as dipping and exposure of the sample in a modifying solution, as well as an alternative method, electrophoretic deposition (EPD), are discussed. Information is provided on the history of the EPD method, its essence, production process, areas of application of this technology, advantages over existing analogues, as well as its varieties. The article considers the promise of using the EPD method to form protective inhibited polymer films on metal surfaces from aqueous solutions of inhibitor formulations consisting of molecules of organosilanes and corrosion inhibitors.

## 1. Introduction

Protection of metal constructions and facilities against corrosion is one of the main tasks in various industries [1,2,3]. One method to reduce of the corrosive activity of process media is the introduction of corrosion inhibitors. According to ISO 8044-2015 [4], corrosion inhibitors are chemical compounds or their compositions, the presence of which in sufficient concentration reduces the corrosion rate of metals without significantly changing in the concentration of corrosive reagents. Corrosion inhibitors can be applied in different forms: volatile [5], contact [6], chamber inhibitors [7], inhibited papers, sleeves, and films [8]. Despite the availability of an extensive range of existing corrosion inhibitors, there is the problem of expanding their assortment by creating new inhibitors or inhibited formulations (INFOR) with higher protective characteristics and lower cost [9,10,11]. When developing new inhibitors or their mixtures, it is necessary to consider the operating environment, the nature of the metal to be protected from corrosion damage, external influences (temperature, pressure, and other factors), etc. [11,12].

In this regard, in recent years, new methods of modifying the metal surface are being used to fight against corrosion, including those based on the use of inhibited formulations consisting of molecules of organosilanes and corrosion inhibitors [13,14,15].

## 2. Options for Protecting of Metal Surfaces from Corrosion Damage by Different Classes of Environmentally Friendly Organic Compounds

### 2.1. Corrosion Inhibitors

Since the slowdown of corrosion processes occurs due to a decrease in the active area of the metal surface and changes in the activation energy of electrode reactions that limit metal corrosion, we can divide inhibitors into three types, namely anodic (affecting the anodic dissolution of the metal surface, including their ability to cause passivation of metal), cathodic (reducing the rate of the cathodic process), and mixed (inhibiting both processes). The protective effect of corrosion inhibitors is based on a change in the state of the metal surface due to their adsorption or on the formation of hard-soluble compounds with metal ions [16,17,18,19,20,21,22,23]. Modern classification of corrosion inhibitors allows their systematization based on their chemical nature; there are oxidative, adsorptive, and co-ordination complex inhibitors, as well as polymeric inhibitors [11,22,24,25,26].

Oxidative corrosion inhibitors have protective properties against many structural metals in a wide pH range and can be used for their protection both in aqueous environments and when operating metals in aggressive atmospheric conditions. The efficient oxidative type inhibitors include salts of chromic acid; however, from an environmental point of view, they are highly toxic, so they had to be replaced by less dangerous ones. For example, molybdates and tungstates have a similar chemical structure to chromates but are not as efficient. In addition, their use is limited by their high cost [27,28,29,30,31,32,33,34,35,36,37]. An alternative to oxidative inhibitors is adsorptive inhibitors.

The protective effect of adsorption type inhibitors is based on the formation of protective layers firmly bonded to the metal surface, isolating the metal surface from the corrosive environment. This effect is due to the state of the metal surface and the charge of the adsorbing particles, as well as their ability to form chemical bonds with the metal or the products of its interaction with the components of the corrosive environment [24,25,26]. As a rule, cation-active inhibitors decelerate active anodic dissolution, i.e., they are effective in the region of potentials lower than the critical passivation potential, or they inhibit cathodic reactions. Anion-active corrosion inhibitors are more effective in preventing local (pitting) corrosion. Complex-forming corrosion inhibitors are very difficult to distinguish from adsorption-type inhibitors as they may not form thick films of complex compounds. Two main groups can be distinguished among the complex-forming inhibitors. The first group includes heterocyclic compounds capable of forming insoluble complexes in aqueous solutions; the second group consists of complexons and metal complexonates. In the first group, the widely used azoles (imidazoles, triazoles, thiazoles) which form on the protected metal surface thin insoluble films of complex compounds with cations of these metals stand out for their effectiveness. The best-known representatives of this class of inhibitors are primarily 1,2,3-benzotriazole (BTA) and its derivatives [38,39,40,41,42,43]. Recent studies show that BTA is widely used not only as a corrosion inhibitor for copper, but also for mild steel and even zinc [38,44,45,46,47,48,49]. In the second group, phosphorus-containing complexones, such as 1-hydroxyethane-1, 1-diphosphonic acid, nitrilotrimethylene phosphonic acid, and ethylenediamine-N,N,N’,N’-tetramethylene phosphonic acid, take the leading place [50,51,52,53,54,55].

Among polymeric inhibitors for neutral media, polyphosphates are the best known [56,57,58]. They are non-toxic, inexpensive, and can inhibit corrosion of steel at low concentrations. Their main disadvantages include the possibility of intensification of corrosion at high concentrations due to the formation of soluble complexes with cations of the protected metal. As early as in the 1970s, in addition to polyphosphates, water-soluble polymers containing COOH^–^ and OH^–^ groups (mainly based on acrylic and maleic acids derivatives), characterized by high hydrolytic stability, started to be used for protection of water systems against scaling [24]. Polymers containing acidic groups are more often used as inhibitors of scaling [59], and cationic polymers for corrosion protection of metals [60]. Another group of corrosion inhibitors of this class is not the polymers themselves, but hydrophilic monomers capable of polymerizing upon adsorption on metal surfaces [59]. A consequence of such polymerization can be a decrease in the solubility of the adsorption layer, an increase in the protective effect, and irreversibility of adsorption.

Even though today a large number of corrosion inhibitors for the majority of structural metals have been effectively tested, there is still a need to expand the range of inhibitors and their compositions with higher protective properties and lower cost. In this vain, researchers aimed to find new corrosion inhibitors or inhibited formulations which do not lose their topicality.

### 2.2. Organosilicon Compounds. Organosilanes

Organosilicon compounds are a class of chemical compounds that contain a bond ≡Si—C≡ in their molecules. The main difference of organosilicon compounds from others is largely due to the low bond strength of bond ≡Si–Si≡ compared to bond ≡C–C≡, and conversely, significantly higher bond strength of bond ≡Si–O–Si≡ than bond ≡C–O–C≡. The values of the bond energies of silicon atoms with each other and with oxygen (213.7 и 444.1 kJ/mol, respectively) compared to those of the bond energies of carbon atoms with each other and with oxygen (347.7 и 358.2 kJ/mol, respectively) confirms this statement. Typically, organosilicon compounds are divided into the following groups: organohalogensilanes, organosilanes, organosiloxanes, and heterocyclic compounds [61,62,63,64]. Let us consider in more detail the class of organosilanes.

Organosilanes are environmentally friendly substances that are not found in nature; they are mainly synthesized from silicon dioxide [65]. The general formula of organosilanes is shown in Figure 1 [62,65,66].

Organosilanes are more prone to condensation reactions, resulting in polysiloxane structure formations that differ markedly in thermal stability from their carbon counterparts [65,66]. Such compounds contain thermally stable siloxane bonds where elements with positive and negative polarization alternate. The presence of a polar substituent in the hydrocarbon radical bonded to a silicon atom leads to an increase in polarity of the polymer molecule and, as a result, an increase in adhesion, mechanical strength, and other properties. In this regard, organosilanes are widely used in the paint industry as adhesion promoters or crosslinking agents that form strong bonds with the overlying layers of coatings and as surface hydrophobisers [65,66,67,68,69,70,71,72]. Thus, the addition of a small amount of organosilanes in the form of 0.1 ÷ 0.5% aqueous solutions improves the adhesion of the polymer matrix to glass fiber [62,73]. The hydrophobicity of the surface after treatment with organosilanes depends on the orientation effect in the organosiloxane layer formed on the surface. The best results are achieved when the siloxane bond is oriented toward the surface and the hydrocarbon radical is oriented from the surface into the external environment [62,74,75].

According to their chemical structure, organosilanes can be divided into two groups: monosilanes (single Si atom) and bis-silanes (two Si atoms). Monosilanes are used as organosilane crosslinking agents, while bis-silanes are used to form crosslinks in silane crosslinking agents [76,77,78]. In the presence of water, in general, the following transformations occur with organosilanes as shown in Figure 2 [62,65,66,74,79,80].

In the first stage, hydrolysis of organosilanes to form silanol occurs in aqueous solution. If a metal sample is placed in an aqueous organosilane solution, the silanol molecules will diffuse to the metal’s oxide-hydroxide surface (stage 2), displacing the adsorbed water molecules from the surface and starting to interact with the hydroxyl groups of the metal surface. As a result of this interaction, hydrogen bonds are formed and enter a condensation reaction with the formation of ≡Si–O–Me bonds on the metal surface layer. In parallel with stages 1 and 2, the adsorbed silanol molecules enter a polycondensation reaction, forming ≡Si–O–Si≡ bridging siloxane bonds resistant to hydrolysis (stage 3). In the last stage, under the influence of temperature, water evaporation and “solidification/cross-linking” of the siloxane structures take place, forming a polymeric siloxane film on the metal surface.

The hydrolysis reaction of organosilanes proceeds spontaneously, without catalysts, but the presence of acids or bases accelerates their hydrolysis [73,79]. It was shown in [80] that the rate of hydrolysis slows down at neutral pH (6.7 ÷ 7.0), and at pH (2.0 ÷ 4.0) the rate of hydrolysis increases by 1000 times. The completeness of hydrolysis is affected by the size of the hydroxyl group in the organosilane molecule; the smaller it is, the higher the rate of hydrolysis. For example, the hydrolysis rate of methoxysilane is 6–10 times higher than that of ethoxysilane [73,80]. In addition, as was shown in [81,82], besides the addition of acids, the hydrolysis reaction rate can be increased if the water–organosilane solution is subjected to ultrasound treatment. Thus, by adjusting the pH of the solution, the rate of hydrolysis, the formation of siloxane bonds to the metal surface, and the polymerization and orientation of the organosilane molecules can be controlled [70,73].

There are several ways to apply organosilanes on different substrates:-By dissolving of organosilane in a solvent mixture of ethanol and water at pH 4.0 to 5.0 (the sample is immersed in the solution and then removed for drying) [77,79,80,81,82];-Vapor phase deposition (in a closed chamber, a tank with organosilane is heated at reduced pressure, forming its vapor, which condenses on a metal surface) [83];-Spin-on deposition (organosilane solution is deposited on a low-speed rotating substrate, followed by washing) [83];-Spray application from aqueous or alcoholic solutions followed by air drying [83].

Some papers [84,85] reported that ornagosilanes such as 3-aminopropyltriethoxysilane, 3-glycidoxypropyltrimethoxysilane, ureidopropyltriethoxysilane, vinyltrimethoxysilane, and functional silanes with the addition of crosslinking silanes such as bis [trimethoxysilylpropyl] amine are quite resistant to corrosive environments. Although organosilanes inhibit corrosive processes on metals, their protective effect is much lower compared to corrosion inhibitors [86,87].

### 2.3. Inhibited Formulations (INFOR) Consisting of Organosilane Molecules and Corrosion Inhibitors

As noted earlier, organosilane films are not always able to protect metals from corrosion damage. The closest analogues of such films are anticorrosive chromate coatings [88,89]; however, their use is limited due to their toxicity. The main disadvantage of siloxane films before chromate coatings is the lack of the so-called “self-healing” ability of siloxanes. Chromate in an aqueous medium is able to diffuse to the damaged area and incorporate into the film, providing recovery of the damaged areas. Numerous studies have been aimed at eliminating this disadvantage of organosilanes by introducing chromium-free corrosion inhibitors into the siloxane film, which contribute to the “healing” of film defects [89].

The use of aqueous solutions of organosilanes with corrosion inhibitors is of interest. Treatment of metal samples in such compositions can lead to a high protective anti-corrosion effect. Films that are formed on metal from INFOR solutions are promising and environmentally safe. Since the compounds used are consumed in small concentrations, the size of the protected product is not changed and the problem associated with its de-conservation is removed [14,15]. Previous studies have shown that INFOR can be used to protect metals in various aggressive environments: in solutions, in atmospheric conditions, etc. Thus, in [90] it was shown that in an aqueous chloride-containing solution, when using the inhibitor composition consisting of organosilane molecules (vinyltrimethoxysilane or diaminsilane) and corrosion inhibitor (1,2,3-benzotriazole), siloxanoazole fragments are formed on the metal surface which provide additional cross-linking of surface adsorbed molecules, thereby increasing the degree of polymerization and, consequently, the density of the surface layers. Moreover, similar inhibitor compositions can be used to form films on metal surfaces to protect metal in various corrosive atmospheres [14,15,90,91]. In this case, the use of carboxylic or phosphonic acids leads to an additional interaction with siloxane groups, which contributes to the formation of a thicker film, and therefore provides improved corrosion resistance [91]. Summarized information from this section is presented in Table 1.

## 3. Methods of Forming Protective Films and Coatings on Metal Surfaces from INFOR Aqueous Solutions

The conventional technique of coating and film deposition is the method of dipping the sample into a solution containing inhibitor composition [11,12,92]. However, this method has disadvantages, the most important one being that the protective properties of the film or coating vary with the exposure time of the sample in the solution (Figure 3) [91].

Figure 2 shows that if the conventional method of a polymer film formation on the metal surface is used, its performance properties (continuity, uniformity, adhesion, and protective properties, etc.) are determined by the duration of exposure of the sample in the INFOR modifying solution. Thus, to form a quality film on the surface of steel and copper, it is necessary to spend at least 9 h [91]. Therefore, it is obviously necessary to optimize the process of obtaining a quality protective film on a metal sample. In this case, electrodeposition can be an alternative method of film formation.

### 3.1. Electrophoretic Deposition (EPD): History of the Method, Its Essence, Advantages, Production Process, and EPD Varieties

The principle of moving particles of matter under the influence of electricity was discovered in 1809 by professors P. I. Strakhov and F. F. Reiss of Moscow University. It was called electrophoresis [93,94,95]. Gradually the study of this phenomenon gained momentum; already in 1917, the first patent for the use of electrophoretic painting was received by General Electric. Beginning in the 1920s, the process began to be used to apply latex rubber [96]. In the 1930s, some of the first patents were obtained describing basic neutralized water-dispersible resins specifically designed for EPD coatings on metals. In the late 1950s, the Ford Motor Company engineering team actively began to develop a methodology for the EPD coating process for cars. The first commercial anodic coating system for automobiles went live in 1963. The first patent for the cathodic product was issued in 1965, and already in 1975, the created technology resulted in the rapid application of cathodic EPD in the automotive industry [95]. In the USSR, this method became popular in the 1980s and found its application in many plants of the automotive, aviation, and machine-building industries [97,98]. Today, the electrophoretic deposition technology accounts for about 70% of the coating of various products. A major part of EPD use is in the automotive industry. It is probably one of the effective methods that can significantly increase the service life of metal structures and products [99,100].

The term EPD includes a wide range of industrial operations: cathodic and anodic electrodeposition as well as electrophoretic coating or electrophoretic painting. During EPD, colloidal particles suspended in a liquid medium migrate under the action of an electric field (electrophoresis) and are deposited on the surface of an oppositely charged electrode [101]. All colloidal particles that can be used to form stable suspensions and that can carry a charge are useful for electrophoretic deposition. Such materials include metals, polymers, pigments, dyes, and ceramics. There are two modes of EPD, namely constant voltage and constant current. The first mode forms thinner coatings than the second one [102,103,104,105,106].

EPD processing has several advantages [99,101,107,108,109]:-The coatings/films applied to the product are continuous and uniform in thickness;-Flms/coatings can be formed on products with complex geometry;-EPD-formed coatings/films have better corrosion and mechanical properties, which ensure a longer service life of the treated product;-Less time is spent per unit compared to immersion/aging samples in modifying solutions;-The technology is applicable to a wide class of materials (metals, ceramics, polymers, etc.);-The process is automated as a rule and does not require large amounts of human resources and special requirements to the operating personnel, which significantly reduces the cost of the films/coatings produced by EPD technology;-Generally, an aqueous solvent is used, reducing the risk of fire in comparison to the solvent-based films/coatings they replace;-Modern electrophoretic materials (varnishes, paints, and other products) are largely more environmentally friendly than materials of other film/coating technologies.

Despite the obvious advantages of EPD, this method has a variety of disadvantages:
-Limited choice of solution compositions because of electrical conductivity and solubility of the components used;-This method allows the application of only a single-layer film/coating;-It is necessary to use expensive equipment, e.g., high-power current sources and drying cabinets of large volume, which leads to an increase in industrial area.

In general terms, the EPD production process can be schematically represented as shown in Figure 4, [109,110,111,112].

The first step is surface preparation. This is usually a process of machining, cleaning/degreasing the metal, and applying pretreatments such as oxidizing or phosphating [109,110]. In the second step, the EPD process begins. The sample is immersed in an electrolyte bath/cell and an electric current is applied through the EPD bath using electrodes. Typically, when electrophoretic films/coatings are deposited, the voltage ranges from 25 to 400 V DC. After deposition, the sample is washed in water to remove the excess undeposited film/coating (step 3). An ultrafilter may be used during the washing process, on which excess deposition material accumulates and then can be returned to the deposition bath; this ensures high material efficiency and reduces the amount of wasting. In the last step, the sample is subjected to a heat treatment, which allows the polymer film or coating to cure. As a result of the thermal treatment, the film, which was porous due to the gas released during the EPD process, spreads out, acquiring a smooth, uniform, and defectless structure [111,112].

In addition, as previously mentioned, EPD is either cathodic or anodic (Figure 5), [99,106,113].

Figure 4 shows that the CPD and APD processes are similar. The difference lies in the polarity of the charge on the surface of the processed product (cathode or anode): in CPD, the surface has a negative electrical charge, and the counter electrodes have a positive charge; in APD, the sample surface is charged with a positive electrical charge, and the counter electrodes have a negative charge [109,113]. In APD, the deposited material contains acidic salts that play the role of charge-carrying groups. These negatively charged anions react with positively charged hydrogen ions (protons), which are formed at the anode as a result of water electrolysis with the conversion of the original acid. A fully protonated acid carries no charge. It is less soluble in water and can precipitate out of the water onto the anode. During CPD, the precipitated material contains basics as charge-carrying groups. If the basic salt was formed by the protonating of the base, such a base will react with the hydroxyl ions produced by the electrolysis of the water to form a neutrally charged base and water. Both types of processes have their advantages and disadvantages. Let us consider the main positives for each method [99,108,109,111,114,115]. The key features of these processes are presented in Table 2.

Thus, when choosing one or another method of electrodeposition, one should consider the nature of the substrate, its pretreatment, the pH of the electrolyte used, as well as the cost of film/coating formation by the chosen method. Consideration of the above factors will greatly facilitate the task of choosing the electrodeposition method for the formation of a high-quality film/coating on the substrate.

### 3.2. Formation of Protective Inhibited Polymer Films on Metals using EPD from INFOR Aqueous Solutions

A review of the current scientific progress on the subject showed that the electrodeposition method, in suspensions containing organosilane, is used either for coatings already pretreated with organosilane or for sol-gel coatings on metal (Figure 6) [116,117,118,119,120,121,122,123].

In the case of pre-silanization, the organosilane is used as an intermediate layer between the metal substrate and the main coating to improve the adhesion properties of the main coating (Figure 6a) [116,117,118,119]. In the case of sol-gels, solutions usually contain one or more organometallic compounds such as zirconium, aluminum, or titanium compounds, one or more organosilanes, and acids, bases, glycols, etc., are used as catalysts (Figure 6b). (Figure 6b) [120,121,122,123]. The addition of organosilanes leads to the formation of denser particles and the sols themselves become more viscous, which, as a result of further electrodeposition, results in virtually defect-free coatings [123,124].

As a result, we can conclude that the very idea of using the EPD method to form polymer inhibited films on metals from aqueous INFOR solutions is brand new. The originality of this proposal lies in the electrolyte used and the film formation method. The electrolyte is an aqueous suspension containing organosilane and corrosion inhibitors in which the main film-forming component is organosilane due to its ability to polycondense. EPD is used to accelerate the deposition of the film on the sample, as well as to orient the siloxane bonds to the metal surface. Since the proposed technology is a “new trend” for such INFORs, there are few works in this area. For example, work [125] reports that using CPD on the surface of tungsten from an aqueous INFOR solution was able to form a polymer film on the surface of tungsten. The paper proposes INFOR composition with acceptable concentrations of the substances used and suggests optimal modes of formation of a quality inhibited polymer film on the surface of tungsten. For example, increasing the duration of cataphoresis contributes to increasing the number of deposited siloxane groups, which complicates the process of further polycondensation (Figure 7) [125].

The figure shows if that the duration of CPD less than 5 min, it leads to the formation of an “island film” on the sample. Thus, it has been experimentally established that, for the formation of a continuous film from aqueous INFOR solutions on the surface of tungsten, the optimal CPD time is 5 min. During this time, the necessary number of siloxane groups is chemisorbed for further formation of a solid film.

### 3.3. The Main Similar Methods of Forming Protective Inhibited Polymer Films on Metals from Aqueous Solutions of INFOR

The proposed EPD technology for the formation of polymer films on metals from aqueous solutions of INFOR can find application in the following areas:-Corrosion protection of metals (preservation of metal products, protection of metals from atmospheric corrosion, etc.) [11,13,126];-Metal surface pretreatment with a primer for its subsequent painting with paint and varnish materials [127,128];-Decorative films [129,130].

Based on the field of application, and the proposed method of polymer films application, it is possible to distinguish the following range of products that can compete with the discussed technology. First of all, these are cataphoresis varnishes, inhibited polymer films/sleeves, as well as water-borne organosoluble paint coatings. The EPD method has several advantages over the listed counterparts; further, we will analyze its positive aspects.

#### 3.3.1. Cataphoresis Varnishes

As a rule, such formulations have a complex, multicomponent, and expensive chemical composition. Usually, the films produced from such suspensions are colorless. However, if there is a need to obtain a color film, then after the application of the polymer film to the substrate, the sample must be further dyed in the pigment toner, which, in turn, leads to an increase in the cost of the product. After the polymer film/coating is applied to the sample, it must be heat cured at T = 120 ÷ 180 ℃ for 20 ÷ 40 min [103,104,129,131,132,133,134,135]. Compared to cataphoresis lacquers, the technology of applying inhibited films of aqueous solutions of INFOR using EPD has the following advantages:-The use of INFOR will lead to a simplification of the electrolyte composition;-Speeding up the drying process of metal products by 25% since, according to preliminary experimental data, it takes about 10 ÷ 15 min for the thermal curing of films;-The cost of an aqueous suspension is significantly lower.

#### 3.3.2. Inhibited Polymer Films/Sleeves

Usually, such films contain a volatile corrosion inhibitor (VCI). The principle of their action is as follows: the product is covered with a film which releases inhibitor vapors, filling all the space inside the package and creating a protective gas environment around the parts or structures. On the metal surface, the VCIs condense and form a monomolecular film that prevents corrosion. Since these compounds are in a gaseous state, they easily penetrate any crevices and cavities, providing protection in the most difficult-to-reach places. This is a major advantage when using inhibited materials containing volatile compounds. Protective action is carried out at a distance of up to 70 cm from the film [8,136,137,138,139].

The main disadvantages of such protection: an auxiliary barrier is required to prevent VCI from escaping from the volume of the protective sleeve/film, and the protected product must be hermetically sealed. This additionally requires the use of special adhesive tape or heat welding [8,139]. Such aspects also increase the cost of the offered products. Compared to inhibited polymer films/sleeves, the films made from INFOR solutions by the EPD method have the following advantages:-The proposed technology does not require additional packaging material;-Economical consumption of the protective material;-Reduction of production labor costs by 2 times.

#### 3.3.3. Water-Borne, Organosoluble Paint Coatings

They are easily available and relatively cheap paintwork materials (paints). Depending on the condition of the polymer binder, waterborne paints are subdivided into water-dispersible and water-soluble. Water-dispersible paintwork materials are suspensions of pigments and fillers in aqueous dispersions of film-forming substances of the synthetic polymers type with the addition of emulsifiers, dispersants, and other auxiliaries. Water-soluble paints, according to the type of film-forming substance, are subdivided into copolymervinyl acetate (the basis is an aqueous dispersion of vinyl acetate copolymers with dibutyl maleate or ethylene); polyvinyl acetate (the basis is a polyvinyl acetate dispersion); butadiene styrene (the basis is latex, which is a copolymer of butadiene with styrene); polyacrylic (the basis is an acrylic dispersion copolymer), etc. [140,141,142,143,144].

Organic solvent paints are paints based organic solvents which evaporate in the drying process. Such paints are divided into two groups: oil and alkyd paints. For oil paints, the binder is oil which dries in the oxidation process. They are linseed oil, linseed oil varnish, oil-saturated alkyd resin, or a mixture of different oils. They are characterized by a fairly long drying process. Alkyd resin is the binder of alkyd materials. As a rule, it is received by boiling vegetable oils, linseed, tall oil, soya oil, etc., together with alcohol and organic acids or acid anhydrides. Like oils, alkyd resins dry out as a result of oxidation [145,146,147,148,149,150].

The main disadvantage of the considered paint-and-lacquer coatings is that they rather quickly sorb moisture, which results in the peeling of the coating from the metall. In addition, some representatives of these materials can be fire-hazardous and toxic [151,152,153].

In comparison with water-soluble and organosoluble paint coatings, the offered method of the formation of polymer films from INFOR solutions possesses the following positive features:-INFOR components are safe;-The polymer inhibited films have a more solid structure that should lead to an increase in the adhesive strength of the film/coating to the metal;-No long preparation of the surface is required;-Formed films can be used as a primer for the following painting of the product.

Summarized information on the comparison of the described methods is presented in Table 3.

## 4. Conclusions

Thus, we can conclude that, for obtaining polymer films/coatings on substrates of different materials, the EPD method is an alternative to traditional methods. This method has several advantages: it optimizes the process of obtaining polymer films/coatings; it is possible to coat samples of different geometric shapes; the formed films/coatings have better performance properties; the risk of ignition of used materials is reduced; environmentally safe substances are used as raw materials. The EPD option is chosen based on the following factors: the nature of the substrate; the method of its pretreatment; the pH of the electrolyte used; and the cost of forming the film/coating by the chosen method. The use of electrophoretic deposition technology is a relatively new and promising method of forming corrosion-resistant polymer films on metal samples from aqueous solutions of inhibitor compositions consisting of molecules of organosilanes and corrosion inhibitors. Cataphoresis varnishes should be considered as direct analogues of this method; however, unlike polymeric inhibited films formed from aqueous INFOR solutions, cataphoresis varnishes do not always have satisfactory adhesion strength and good protective capacity.

## Figures and Tables

**Figure 1 materials-16-00019-f001:**
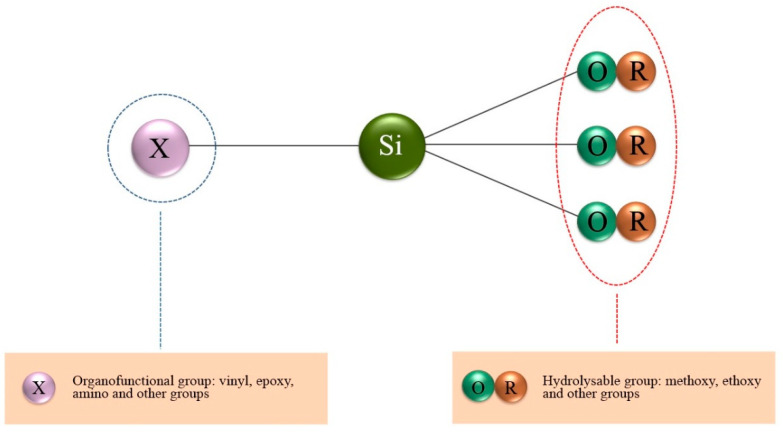
The general formula of organosilanes.

**Figure 2 materials-16-00019-f002:**
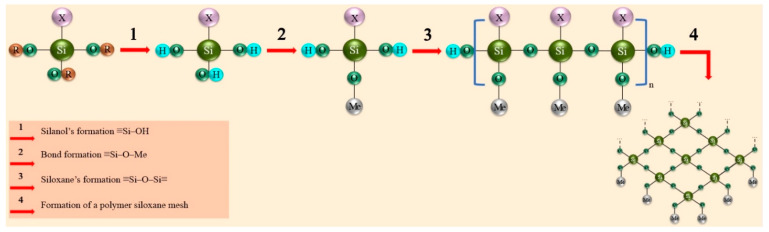
The reactions that result in the formation of a polymer siloxane film on a metal surface.

**Figure 3 materials-16-00019-f003:**
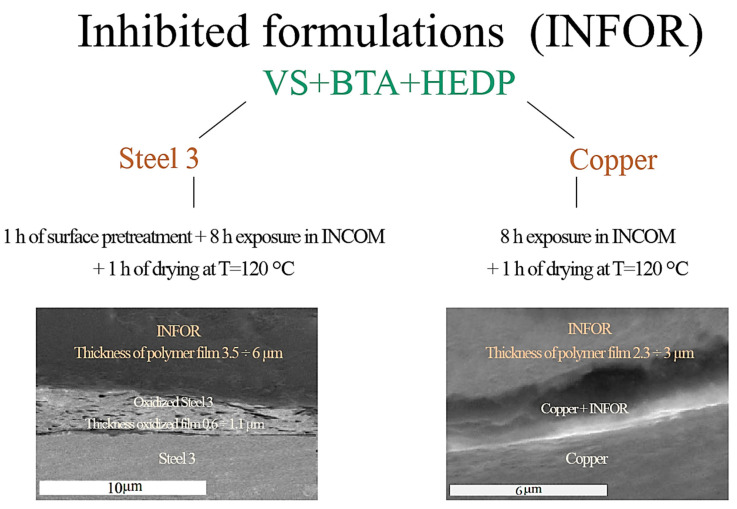
SEM images of cross-sections of polymer films on steel 3 and copper.

**Figure 4 materials-16-00019-f004:**
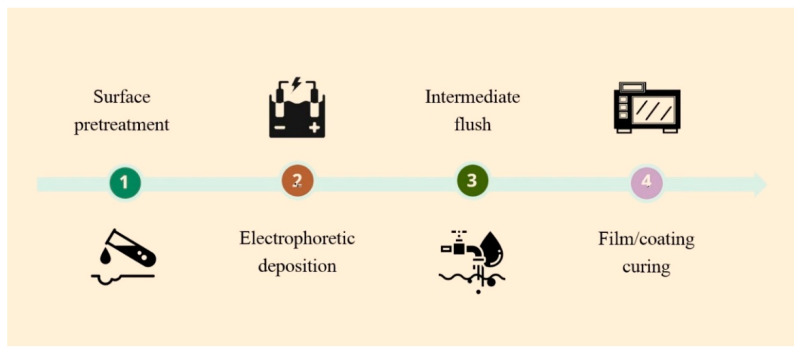
The production process of electrophoretic deposition.

**Figure 5 materials-16-00019-f005:**
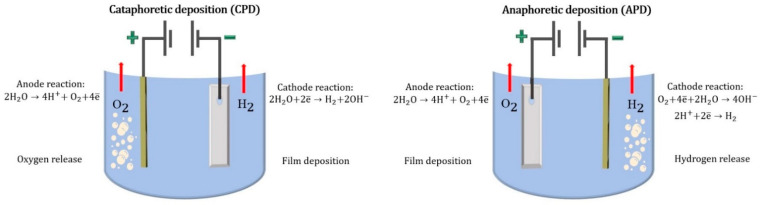
The processes taking place during cataphoretic and anaphoretic deposition.

**Figure 6 materials-16-00019-f006:**
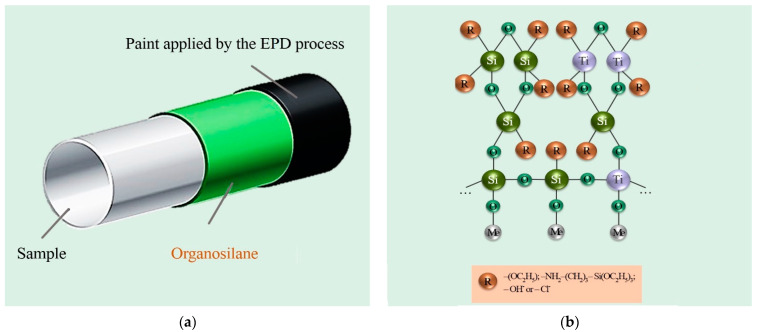
Examples of the use of organosilanes in EPD: (**a**) preliminary silanization of the metal sample surface; (**b**) application of organosilanes in colloidal sol-gels.

**Figure 7 materials-16-00019-f007:**
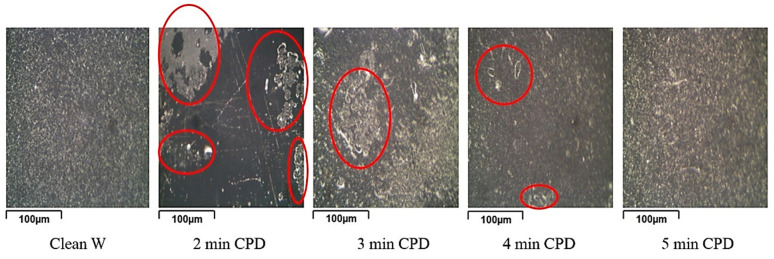
Appearance of tungsten samples subjected to CPD in INFOR aqueous solution depending on the duration of CPD. Uncovered areas of the sample are marked with the red circle.

**Table 1 materials-16-00019-t001:** Summary of protection of metals by organic compounds.

Criteria	Corrosion Inhibitors	Organosilanes	INFOR
Type of protective action	Oxidative [11,22,27,28,29,30,31,32,33,34,35,36,37] Adsorptive [24,25,26,30,31,32,33,34,35,36,37] Complex forming [38,39,40,41,42,43,44,45,46,47,48,49] Polymeric [56,57,58,59,60]	Film forming (isolaing) [79,80,81,82,83,84,85,86,87]	Isolating [88,89,90,91]
Healing effect	High for chromates [24,25,88,89]	Moderate [61,62,63,64,65,66,67]	High [14,15,90,91]
Application form	Volatile [4,5,11,12,13] Contact [6,16,17,18,19,20,21,22,23,24,25,26,27,28,29,30,31,32,33,34,35,36,37,38,40,41,48,49] Chamber [7,9,10,39,42] Inhibited papers, sleeves [8]	Ethanol-water solution, applied by immersion [77,79,80,81,82], vapor phase, spin-on, spray [83]	Water solution with organosilane and contact inhibitor [14,15,90,91]

**Table 2 materials-16-00019-t002:** The key features of the APD and CPD processes.

Method	Feature
CPD	The films/coatings produced by this method have higher protective properties. However, this effect may be due to the cross-linking chemistry of the raw material (polymer) used rather than to the electrode on which the film/coating is deposited; The product can be designed with less current density due to the higher throw power of the medium; The oxidation process takes place at the anode, so staining and other problems that could result from the oxidation of the metal substrate are eliminated.
APD	Compared to CPD, APD is less expensive; Less sensitivity to changes in substrate quality; The substrate is not exposed to strong alkaline attack which can dissolve phosphate, oxide, and other coatings used as substrate pretreatment; The anodic process avoids hydrogen embrittlement, which can occur during the cathodic process, due to hydrogen ion discharge.

**Table 3 materials-16-00019-t003:** Comparison of the key features of the protective film formation methods.

Method/Technique	Conveniences	Limitations
Organosilanes films by dipping in INFOR	No complex equipment is required Environmentally safe	Metal surface needs to be pre-treated Long process of coating formation
	[14,15,90,91,125]	
Organosilanes films by EPD of INFOR	Accelerated coating formation process Complex shapes can be coated Environmentally safe	Requires more expensive equipment
	[125,126,127,128,129,130]	
Cataphoresis varnishes	Uniform coating Complex shapes can be coated Relatively high wear resistance	Complex solution composition Requires more expensive Equipment
	[103,104,131,132,133,134,135]	
Inhibited sleeves/films	Easy to apply Relatively cheap	Need to be sealed due to a danger of inhibitor volatilization
	[8,136,137,138,139]	
Paint coatings	Proven process of coating formation Relatively cheap	Sorb moisture, toxic Relatively high consumption of paint material
	[140,141,142,143,144,145,146,147,148,149,150,151,152,153]	

## Data Availability

Not applicable.

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
