# Peer review of "Application of Electrophoretic Deposition as an Advanced Technique of Inhibited Polymer Films Formation on Metals from Environmentally Safe Aqueous Solutions of Inhibited Formulations"

_materials, 2022, doi:10.3390/ma16010019_

Round 1

Reviewer 1 Report

The authors analyzed polymer films formed from aqueous solutions of organosilanes, corrosion inhibitors and their compositions. They try to introduce the EPD technique. However,  some issues should be addressed.

1. The title is too long and it is hard to read.

2. What's the disadvantage of EPD?

3.The authors should compared other techniques with EPD.

4. The language should be improved.

Reviewer 2 Report

In this manuscript, the authors presented a review of methods of polymer film formation and proposed the electrophoretic deposition EPD as an alternative method to form these polymer film coatings. This method is compared to traditional methods such as dipping and exposing the samples in a modifier solution. It is concluded that the EPD method can coat samples of different geometrical shapes with better performance and the risk of ignition of the materials used is reduced.

In the reviewer opinion, the paper can be recommended for publication in materials journal after addressing the following comments:

-  All the figures presented in the present manuscript are without any references. It is better to add the reference of each figure.

- It is recommended to replace in the title scheme 1 by figure 2.

- It is recommended that tables be added in each section to analyze the results and contributions of the various references in order to summarize these works and better see their contributions to the field.

- Figure 3: If possible, it is recommended to add a real figure of the process to better understand the EPD method.

- The comparison between APD and CPD presented at the end of section 3.1 can be presented in a table for greater clarity.

- Figure 6. Red circles are drawn in the figure to present the different defects. It is recommended to specify on the text the type of these defects and what is the cause.

- At the end of the manuscript, before the conclusion section, it is recommended to add a table comparing the different methods cited in the text, presenting the advantages and disadvantages of each technique and highlighting the alternatives proposed in the manuscript.

Round 2

Reviewer 1 Report

Accept